# Sparse Precision Matrix Estimation with Calibration

**Tuo Zhao**
Department of Computer Science
Johns Hopkins University

**Han Liu**
Department of Operations Research and Financial Engineering
Princeton University

## Abstract

We propose a semiparametric method for estimating sparse precision matrix of high dimensional elliptical distribution. The proposed method calibrates regularizations when estimating each column of the precision matrix. Thus it not only is asymptotically tuning free, but also achieves an improved finite sample performance. Theoretically, we prove that the proposed method achieves the parametric rates of convergence in both parameter estimation and model selection. We present numerical results on both simulated and real datasets to support our theory and illustrate the effectiveness of the proposed estimator.

## 1 Introduction

We study the precision matrix estimation problem: let $\boldsymbol{X} = (X_1, ..., X_d)^T$ be a $d$-dimensional random vector following some distribution with mean $\boldsymbol{\mu} \in \mathbb{R}^d$ and covariance matrix $\boldsymbol{\Sigma} \in \mathbb{R}^{d \times d}$, where $\boldsymbol{\Sigma}_{kj} = \mathbb{E}X_k X_j - \mathbb{E}X_k \mathbb{E}X_j$. We want to estimate $\boldsymbol{\Omega} = \boldsymbol{\Sigma}^{-1}$ from $n$ independent observations. To make the estimation manageable in high dimensions ($d/n \to \infty$), we assume that $\boldsymbol{\Omega}$ is sparse. That is, many off-diagonal entries of $\boldsymbol{\Omega}$ are zeros.

Existing literature in machine learning and statistics usually assumes that $\boldsymbol{X}$ follows a multivariate Gaussian distribution, i.e., $\boldsymbol{X} \sim N(0, \boldsymbol{\Sigma})$. Such a distributional assumption naturally connects sparse precision matrices with Gaussian graphical models (Dempster, 1972), and has motivated numerous applications (Lauritzen, 1996). To estimate sparse precision matrices for Gaussian distributions, many methods in the past decade have been proposed based on the sample covariance estimator. Let $\boldsymbol{x}_1, ..., \boldsymbol{x}_n \in \mathbb{R}^d$ be $n$ independent observations of $\boldsymbol{X}$, the sample covariance estimator is defined as

$$\mathbf{S} = \frac{1}{n} \sum_{i=1}^{n} (\boldsymbol{x}_i - \bar{\boldsymbol{x}})(\boldsymbol{x}_i - \bar{\boldsymbol{x}})^T \text{ with } \bar{\boldsymbol{x}} = \frac{1}{n} \sum_{i=1}^{n} \boldsymbol{x}_i. \tag{1.1}$$

Banerjee et al. (2008); Yuan and Lin (2007); Friedman et al. (2008) take advantage of the Gaussian likelihood, and propose the graphic lasso (GLASSO) estimator by solving

$$\widehat{\boldsymbol{\Omega}} = \underset{\boldsymbol{\Omega}}{\operatorname{argmin}} - \log |\boldsymbol{\Omega}| + \operatorname{tr}(\mathbf{S}\boldsymbol{\Omega}) + \lambda \sum_{j,k} |\boldsymbol{\Omega}_{kj}|,$$

where $\lambda > 0$ is the regularization parameter. Scalable software packages for GLASSO have been developed, such as `huge` (Zhao et al., 2012).

In contrast, Cai et al. (2011); Yuan (2010) adopt the pseudo-likelihood approach to estimate the precision matrix. Their estimators follow a column-by-column estimation scheme, and possess better

theoretical properties. More specifically, given a matrix $\mathbf{A} \in \mathbb{R}^{d \times d}$, let $\mathbf{A}_{*j} = (\mathbf{A}_{1j}, ..., \mathbf{A}_{dj})^T$ denote the $j^{\text{th}}$ column of $\mathbf{A}$, $||\mathbf{A}_{*j}||_1 = \sum_k |\mathbf{A}_{kj}|$ and $||\mathbf{A}_{*j}||_\infty = \max_k |\mathbf{A}_{kj}|$, Cai et al. (2011) obtain the CLIME estimator by solving

$$\widehat{\mathbf{\Omega}}_{*j} = \underset{\mathbf{\Omega}_{*j}}{\operatorname{argmin}} \ ||\mathbf{\Omega}_{*j}||_1 \ \text{ s.t. } ||\mathbf{S}\mathbf{\Omega}_{*j} - \mathbf{I}_{*j}||_\infty \leq \lambda, \ \ \forall \, j = 1, ..., d. \tag{1.2}$$

Computationally, (1.2) can be reformulated and solved by general linear program solvers. Theoretically, let $||\mathbf{A}||_1 = \max_j ||\mathbf{A}_{*j}||_1$ be the matrix $\ell_1$ norm of $\mathbf{A}$, and $||\mathbf{A}||_2$ be the largest singular value of $\mathbf{A}$, (i.e., the spectral norm of $\mathbf{A}$), Cai et al. (2011) show that if we choose

$$\lambda \asymp ||\mathbf{\Omega}||_1 \sqrt{\frac{\log d}{n}}, \tag{1.3}$$

the CLIME estimator achieves the following rates of convergence under the spectral norm,

$$||\widehat{\mathbf{\Omega}} - \mathbf{\Omega}||_2^2 = O_P \left( ||\mathbf{\Omega}||_1^{4-4q} s^2 \left( \frac{\log d}{n} \right)^{1-q} \right), \tag{1.4}$$

where $q \in [0, 1)$ and $s = \max_j \sum_k |\mathbf{\Omega}_{kj}|^q$.

Despite of these good properties, the CLIME estimator in (1.2) has three drawbacks: (1) The theoretical justification heavily relies on the subgaussian tail assumption. When this assumption is violated, the inference can be unreliable; (2) All columns are estimated using the same regularization parameter, even though these columns may have different sparseness. As a result, more estimation bias is introduced to the denser columns to compensate the sparser columns. In another word, the estimation is not calibrated (Liu et al., 2013); (3) The selected regularization parameter in (1.3) involves the unknown quantity $||\mathbf{\Omega}||_1$. Thus we have to carefully tune the regularization parameter over a refined grid of potential values in order to get a good finite-sample performance. To overcome the above three drawbacks, we propose a new sparse precision matrix estimation method, named EPIC (Estimating Precision mIatrix with Calibration).

To relax the Gaussian assumption, our EPIC method adopts an ensemble of the transformed Kendall's tau estimator and Catoni's M-estimator (Kruskal, 1958; Catoni, 2012). Such a semiparametric combination makes EPIC applicable to the elliptical distribution family. The elliptical family (Cambanis et al., 1981; Fang et al., 1990) contains many multivariate distributions such as Gaussian, multivariate $t$-distribution, Kotz distribution, multivariate Laplace, Pearson type II and VII distributions. Many of these distributions do not have subgaussian tails, thus the commonly used sample covariance-based sparse precision matrix estimators often fail miserably.

Moreover, our EPIC method adopts a calibration framework proposed in Gautier and Tsybakov (2011), which reduces the estimation bias by calibrating the regularization for each column. Meanwhile, the optimal regularization parameter selection under such a calibration framework does not require any prior knowledge of unknown quantities (Belloni et al., 2011). Thus our EPIC estimator is asymptotically tuning free (Liu and Wang, 2012). Our theoretical analysis shows that if the underlying distribution has a finite fourth moment, the EPIC estimator achieves the same rates of convergence as (1.4). Numerical experiments on both simulated and real datasets show that EPIC outperforms existing precision matrix estimation methods.

## 2 Background

We first introduce some notations used throughout this paper. Given a vector $\boldsymbol{v} = (v_1, \ldots, v_d)^T \in \mathbb{R}^d$, we define the following vector norms:

$$||\boldsymbol{v}||_1 = \sum_j |v_j|, \ ||\boldsymbol{v}||_2^2 = \sum_j v_j^2, \ ||\boldsymbol{v}||_\infty = \max_j |v_j|.$$

Given a matrix $\mathbf{A} \in \mathbb{R}^{d \times d}$, we use $\mathbf{A}_{*j} = (\mathbf{A}_{1j}, ..., \mathbf{A}_{dj})^T$ to denote the $j^{\text{th}}$ column of $\mathbf{A}$. We define the following matrix norms:

$$||\mathbf{A}||_1 = \max_j ||\mathbf{A}_{*j}||_1, ||\mathbf{A}||_2 = \max_j \psi_j(\mathbf{A}), \ ||\mathbf{A}||_{\text{F}}^2 = \sum_{k,j} \mathbf{A}_{kj}^2, \ ||\mathbf{A}||_{\max} = \max_{k,j} |\mathbf{A}_{kj}|,$$

where $\psi_j(\mathbf{A})$'s are all singular values of $\mathbf{A}$.

We then briefly review the elliptical family. As a generalization of the Gaussian distribution, it has the following definition.

**Definition 2.1** (Fang et al. (1990)). *Given $\boldsymbol{\mu} \in \mathbb{R}^d$ and $\boldsymbol{\Xi} \in \mathbb{R}^{d \times d}$, where $\boldsymbol{\Xi} \succeq 0$ and $\mathrm{rank}(\boldsymbol{\Xi}) = r \le d$, we say that a d-dimensional random vector $\boldsymbol{X} = (X_1, ..., X)^T$ follows an elliptical distribution with parameter $\boldsymbol{\mu}$, $\boldsymbol{\Xi}$, and $\beta$, if $\boldsymbol{X}$ has a stochastic representation*

$$\boldsymbol{X} \overset{d}{=} \boldsymbol{\mu} + \beta \mathbf{B} \boldsymbol{U},$$

*such that $\beta \ge 0$ is a continuous random variable independent of $\boldsymbol{U}$, $\boldsymbol{U} \in \mathbb{S}^{r-1}$ is uniformly distributed in the unit sphere in $\mathbb{R}^r$, and $\boldsymbol{\Xi} = \mathbf{B}\mathbf{B}^T$.*

Since we are interested in the precision matrix estimation, we assume that $\max_j \mathbb{E} X_j^2$ is finite. Note that the stochastic representation in Definition 2.1 is not unique, and existing literature usually imposes the constraint $\max_j \boldsymbol{\Xi}_{jj} = 1$ to make the distribution identifiable (Fang et al., 1990). However, such a constraint does not necessarily make $\boldsymbol{\Xi}$ the covariance matrix. Here we present an alternative representation as follows.

**Proposition 2.2.** *If $\boldsymbol{X}$ has the stochastic representation $\boldsymbol{X} = \boldsymbol{\mu} + \beta \mathbf{B} \boldsymbol{U}$ as in Definition 2.1, given $\boldsymbol{\Xi} = \mathbf{B}\mathbf{B}^T$, $\mathrm{rank}(\boldsymbol{\Xi}) = r$, and $\mathbb{E}(\xi^2) = \alpha < \infty$, $\boldsymbol{X}$ can be rewritten as $\boldsymbol{X} = \boldsymbol{\mu} + \xi \mathbf{A} \boldsymbol{U}$, where $\xi = \beta \sqrt{r/\alpha}$, $\mathbf{A} = \mathbf{B} \sqrt{\alpha/r}$ and $\boldsymbol{\Sigma} = \mathbf{A}\mathbf{A}^T$. Moreover we have*

$$\mathbb{E}(\xi^2) = r, \ \mathbb{E}(\boldsymbol{X}) = \boldsymbol{\mu}, \ and \ \mathrm{Cov}(\boldsymbol{X}) = \boldsymbol{\Sigma}.$$

After the reparameterization in Proposition 2.2, the distribution is identifiable with $\boldsymbol{\Sigma}$ defined as the conventional covariance matrix.

**Remark 2.3.** *$\boldsymbol{\Sigma}$ has the decomposition $\boldsymbol{\Sigma} = \boldsymbol{\Theta} \mathbf{Z} \boldsymbol{\Theta}$, where $\mathbf{Z}$ is the Pearson correlation matrix, and $\boldsymbol{\Theta} = \mathrm{diag}(\theta_1, ..., \theta_d)$ with $\theta_j$ as the standard deviation of $X_j$. Since $\boldsymbol{\Theta}$ is a diagonal matrix, the precision $\boldsymbol{\Omega}$ also has a similar decomposition $\boldsymbol{\Omega} = \boldsymbol{\Theta}^{-1} \boldsymbol{\Gamma} \boldsymbol{\Theta}^{-1}$, where $\boldsymbol{\Gamma} = \mathbf{Z}^{-1}$ is the inverse correlation matrix.*

## 3 Method

We propose a three-step method: (1) We first use the transformed Kendall's tau estimator and Catoni's M-estimator to obtain $\widehat{\mathbf{Z}}$ and $\widehat{\boldsymbol{\Theta}}$ respectively. (2) We then plug $\widehat{\mathbf{Z}}$ into the calibrated inverse correlation matrix estimation to obtain $\widehat{\boldsymbol{\Gamma}}$. (3) At last, we assemble $\widehat{\boldsymbol{\Gamma}}$ and $\widehat{\boldsymbol{\Theta}}$ to obtain $\widehat{\boldsymbol{\Omega}}$.

### 3.1 Correlation Matrix and Standard Deviation Estimation

To estimate $\mathbf{Z}$, we adopt the transformed Kendall's tau estimator proposed in Liu et al. (2012). Given $n$ independent observations, $\boldsymbol{x}_1, ..., \boldsymbol{x}_n$, where $\boldsymbol{x}_i = (x_{i1}, ..., x_{id})^T$, we calculate the Kendall's statistic by

$$\widehat{\tau}_{kj} = \begin{cases} \dfrac{2}{n(n-1)} \sum_{i<i'} \mathrm{sign}\left( (x_{ij} - x_{i'j})(x_{ik} - x_{i'k}) \right) & \text{if } j \ne k; \\ 1 & \text{otherwise.} \end{cases}$$

After a simple transformation, we obtain a correlation matrix estimator $\widehat{\mathbf{Z}} = [\widehat{\mathbf{Z}}_{kj}] = \left[ \sin\left( \frac{\pi}{2} \widehat{\tau}_{kj} \right) \right]$ (Liu et al., 2012; Zhao et al., 2013).

To estimate $\boldsymbol{\Theta} = \mathrm{diag}(\theta_1, ..., \theta_d)$, we adopt the Catoni's M-estimator proposed in Catoni (2012). We define

$$\psi(t) = \mathrm{sign}(t) \log(1 + |t| + t^2/2),$$

where $\mathrm{sign}(0) = 0$. Let $\widehat{m}_j$ be the estimator of $\mathbb{E} X_j^2$, we solve

$$\sum_{i=1}^n \psi \left( (x_{ij} - \widehat{\mu}_j) \sqrt{\frac{2}{n K_{\max}}} \right) = 0, \sum_{i=1}^n \psi \left( (x_{ij}^2 - \widehat{m}_j) \sqrt{\frac{2}{n K_{\max}}} \right) = 0.$$

where $K_{\max}$ is an upper bound of $\max_j \mathrm{Var}(X_j)$ and $\max_j \mathrm{Var}(X_j^2)$. Since $\psi(t)$ is a strictly increasing function in $t$, $\widehat{\mu}_j$ and $\widehat{m}_j$ are unique and can be obtained by the efficient Newton-Raphson method (Stoer et al., 1993). Then we can obtain $\widehat{\theta}_j$ using $\widehat{\theta}_j = \sqrt{\widehat{m}_j - \widehat{\mu}_j^2}$.

## 3.2 Calibrated Inverse Correlation Matrix Estimation

We plugin $\widehat{\mathbf{Z}}$ into the following convex program,

$$(\widehat{\boldsymbol{\Gamma}}_{*j}, \widehat{\tau}_j) = \operatorname*{argmin}_{\boldsymbol{\Gamma}_{*j}, \tau_j} \; ||\boldsymbol{\Gamma}_{*j}||_1 + c\tau_j$$

$$\text{s.t.} \quad ||\widehat{\mathbf{Z}}\boldsymbol{\Gamma}_{*j} - \mathbf{I}_{*j}||_\infty \leq \lambda\tau_j, \; ||\boldsymbol{\Gamma}_{*j}||_1 \leq \tau_j, \; \forall \, j = 1, ..., d. \tag{3.1}$$

where $c$ can be an arbitrary constant (e.g. $c = 0.5$). $\tau_j$ works as an auxiliary variable to calibrate the regularization.

**Remark 3.1.** *If we know $\tau_j = ||\boldsymbol{\Omega}_{*j}||_1$ in advance, we can consider a simple variant of the CLIME estimator,*

$$\widehat{\boldsymbol{\Omega}}_{*j} = \operatorname*{argmin}_{\boldsymbol{\Omega}_{*j}} \; ||\boldsymbol{\Omega}_{*j}||_1$$

$$\text{s.t.} \quad ||\mathbf{S}\boldsymbol{\Omega}_{*j} - \mathbf{I}_{*j}||_\infty \leq \lambda\tau_j, \; \forall \, j = 1, ..., d.$$

*Since we do not have any prior knowledge of $\tau_j's$, we consider the following replacement*

$$(\widehat{\boldsymbol{\Gamma}}_{*j}, \widehat{\tau}_j) = \operatorname*{argmin}_{\boldsymbol{\Gamma}_{*j}, \tau_j} \; ||\boldsymbol{\Omega}_{*j}||_1 \tag{3.2}$$

$$\text{s.t.} \quad ||\mathbf{S}\boldsymbol{\Omega}_{*j} - \mathbf{I}_{*j}||_\infty \leq \lambda\tau_j, \; \tau_j = ||\boldsymbol{\Omega}_{*j}||_1 \; \forall \, j = 1, ..., d.$$

*As can be seen, (3.2) is nonconvex due to the constraint $\tau_j = ||\boldsymbol{\Omega}_{*j}||_1$. Thus no global optimum can be guaranteed in polynomial time.*

From a computational perspective, (3.1) can be viewed as a convex relaxation of (3.2). Both the objective function and the constraint in (3.1) contain $\tau_j$ to prevent from choosing $\tau_j$ either too large or too small. Due to the complementary slackness, (3.1) eventually encourages the regularization to be proportional to the $\ell_1$ norm of each column (weak sparseness). Therefore the estimation is calibrated.

By introducing the decomposition $\boldsymbol{\Gamma}_{*j} = \boldsymbol{\Gamma}_{*j}^+ - \boldsymbol{\Gamma}_{*j}^-$ with $\boldsymbol{\Gamma}_{*j}^+, \boldsymbol{\Gamma}_{*j}^- \geq \mathbf{0}$, we can reformulate (3.1) as a linear program as follows,

$$(\widehat{\boldsymbol{\Gamma}}_{*j}^+, \widehat{\boldsymbol{\Gamma}}_{*j}^-, \widehat{\tau}_j) = \operatorname*{argmin}_{\boldsymbol{\Gamma}_{*j}^+, \boldsymbol{\Gamma}_{*j}^-, \tau_j} \; \mathbf{1}^T\boldsymbol{\Gamma}_{*j}^+ + \mathbf{1}^T\boldsymbol{\Gamma}_{*j}^- + c\tau_j \tag{3.3}$$

$$\text{subjected to} \quad \begin{bmatrix} \widehat{\mathbf{Z}} & -\widehat{\mathbf{Z}} & -\boldsymbol{\lambda} \\ -\widehat{\mathbf{Z}} & \widehat{\mathbf{Z}} & -\boldsymbol{\lambda} \\ \mathbf{1}^T & \mathbf{1}^T & -1 \end{bmatrix} \begin{bmatrix} \boldsymbol{\Gamma}_{*j}^+ \\ \boldsymbol{\Gamma}_{*j}^- \\ \tau_j \end{bmatrix} \leq \begin{bmatrix} \mathbf{I}_{*j} \\ -\mathbf{I}_{*j} \\ 0 \end{bmatrix},$$

$$\boldsymbol{\Gamma}_{*j}^+ \geq \mathbf{0}, \; \boldsymbol{\Gamma}_{*j}^- \geq \mathbf{0}, \; \tau_j \geq 0,$$

where $\boldsymbol{\lambda} = (\lambda, ..., \lambda)^T \in \mathbb{R}^d$. (3.3) can be solved by existing linear program solvers, and further accelerated by the parallel computing techniques.

**Remark 3.2.** *Though (3.1) looks more complicated than (1.2), it is not necessarily more computationally difficult. After the reparameterization, (3.3) contains $2d + 1$ parameters to optimize, which is of a similar scale to the linear program formulation as the CLIME method in Cai et al. (2011).*

Our EPIC method does not guarantee the symmetry of the estimator $\widehat{\boldsymbol{\Gamma}}$. Thus we need the following symmetrization methods to obtain the symmetric replacement $\widetilde{\Gamma}$.

$$\widetilde{\boldsymbol{\Gamma}}_{kj} = \widehat{\boldsymbol{\Gamma}}_{kj}I(|\widehat{\boldsymbol{\Gamma}}_{kj}| \leq \widehat{\boldsymbol{\Gamma}}_{jk}) + \widehat{\boldsymbol{\Gamma}}_{jk}I(|\widehat{\boldsymbol{\Gamma}}_{kj}| > \widehat{\boldsymbol{\Gamma}}_{jk}).$$

## 3.3 Precision Matrix Estimation

Once we obtain the estimated inverse correlation matrix $\widetilde{\boldsymbol{\Gamma}}$, we can recover the precision matrix estimator by the ensemble rule,

$$\widehat{\boldsymbol{\Omega}} = \widehat{\boldsymbol{\Theta}}^{-1}\widetilde{\boldsymbol{\Gamma}}\widehat{\boldsymbol{\Theta}}^{-1}.$$

**Remark 3.3.** *A possible alternative is to directly estimate $\boldsymbol{\Omega}$ by plugging a covariance estimator*

$$\widehat{\mathbf{S}} = \widehat{\boldsymbol{\Theta}}\widehat{\mathbf{Z}}\widehat{\boldsymbol{\Theta}} \tag{3.4}$$

*into (3.1) instead of $\widehat{\mathbf{Z}}$, but this direct estimation procedure makes the regularization parameter selection sensitive to $\mathrm{Var}(X_j^2)$.*

# 4 Statistical Properties

In this section, we study statistical properties of the EPIC estimator. We define the following class of sparse symmetric matrices,

$$\mathcal{U}_q(s, M) = \left\{ \mathbf{\Gamma} \in \mathbb{R}^{d \times d} \,\Big|\, \mathbf{\Gamma} \succ 0, \ \mathbf{\Gamma} = \mathbf{\Gamma}^T, \ \max_j \sum_k |\mathbf{\Gamma}_{kj}|^q \leq s, \ ||\mathbf{\Gamma}||_1 \leq M \right\},$$

where $q \in [0, 1)$ and $(s, d, M)$ can scale with the sample size $n$. We also impose the following additional conditions:

(A.1) $\mathbf{\Gamma} \in \mathcal{U}_q(s, M)$
(A.2) $\max_j |\mu_j| \leq \mu_{\max}$, $\max_j \theta_j \leq \theta_{\max}$, $\min_j \theta_j \geq \theta_{\min}$
(A.3) $\max_j \mathbb{E} X_j^4 \leq K$

where $\mu_{\max}$, $K$, $\theta_{\max}$, and $\theta_{\min}$ are constants.

Before we proceed with our main results, we first present the following key lemma.

**Lemma 4.1.** *Suppose that $\mathbf{X}$ follows an elliptical distribution with mean $\boldsymbol{\mu}$, and covariance $\mathbf{\Sigma} = \mathbf{\Theta} \mathbf{Z} \mathbf{\Theta}$. Assume that (A.1)-(A.3) hold, given the transformed Kendall's tau estimator and Catoni's M-estimator defined in Section 3, there exist universal constants $\kappa_1$ and $\kappa_2$ such that for large enough $n$,*

$$\mathbb{P}\left( \max_j |\widehat{\theta}_j^{-1} - \theta_j^{-1}| \leq \kappa_2 \sqrt{\frac{\log d}{n}} \right) \geq 1 - \frac{2}{d^3},$$

$$\mathbb{P}\left( \max_{j,k} |\widehat{\mathbf{Z}}_{kj} - \mathbf{Z}_{kj}| \leq \kappa_1 \sqrt{\frac{\log d}{n}} \right) \geq 1 - \frac{1}{d^3}.$$

Lemma 4.1 implies that both transformed Kendall's tau estimator and Catoni's M-estimator possess good concentration properties, which enable us to obtain a consistent estimator of $\mathbf{\Omega}$.

The next theorem presents the rates of convergence under the matrix $\ell_1$ norm, spectral norm, Frobenius norm, and max norm.

**Theorem 4.2.** *Suppose that $\mathbf{X}$ follows an elliptical distribution. Assume (A.1)-(A.3) hold, there exist universal constants $C_1$, $C_2$, and $C_3$ such that by taking*

$$\lambda = \kappa_1 \sqrt{\frac{\log d}{n}}, \tag{4.1}$$

*for large enough $n$ and $p = 1, 2$, we have*

$$||\widehat{\mathbf{\Omega}} - \mathbf{\Omega}||_p^2 \leq C_1 M^{4-4q} s^2 \left( \frac{\log d}{n} \right)^{1-q},$$

$$\frac{1}{d} ||\widehat{\mathbf{\Omega}} - \mathbf{\Omega}||_F^2 \leq C_2 M^{4-2q} s \left( \frac{\log d}{n} \right)^{1-q/2},$$

$$||\widehat{\mathbf{\Omega}} - \mathbf{\Omega}||_{\max} \leq C_3 M^2 \sqrt{\frac{\log d}{n}},$$

*with probability at least $1 - 3\exp(-3\log d)$. Moreover, when the exact sparsity holds (i.e., $q = 0$), let $E = \{(k, j) \mid \mathbf{\Omega}_{kj} \neq 0\}$, and $\widehat{E} = \{(k, j) \mid \widehat{\mathbf{\Omega}}_{kj} \neq 0\}$, then we have $\mathbb{P}\left( E \subseteq \widehat{E} \right) \to 1$, if there exists a large enough constant $C_4$ such that*

$$\min_{(k,j) \in E} |\mathbf{\Omega}_{kj}| \geq C_4 M^2 \sqrt{\frac{\log d}{n}}.$$

The rates of convergence in Theorem 4.2 are comparable to those in Cai et al. (2011).

**Remark 4.3.** *The selected tuning parameter $\lambda$ in (4.1) does not involve any unknown quantity. Therefore our EPIC method is asymptotically tuning free.*

## 5    Numerical Simulations

In this section, we compare the proposed ALCE method with other methods including

(1) GLASSO.RC : GLASSO + $\widehat{\mathbf{S}}$ defined in (3.4) as the input covariance matrix

(2) CLIME.RC: CLIME + $\widehat{\mathbf{S}}$ as the input covariance matrix

(3) CLIME.SM: CLIME + $\mathbf{S}$ defined in (1.1) as the input covariance matrix

We consider three different settings for the comparison: (1) $d = 100$; (2) $d = 200$; (3) $d = 400$. We adopt the following three graph generation schemes, as illustrated in Figure 1, to obtain precision matrices.

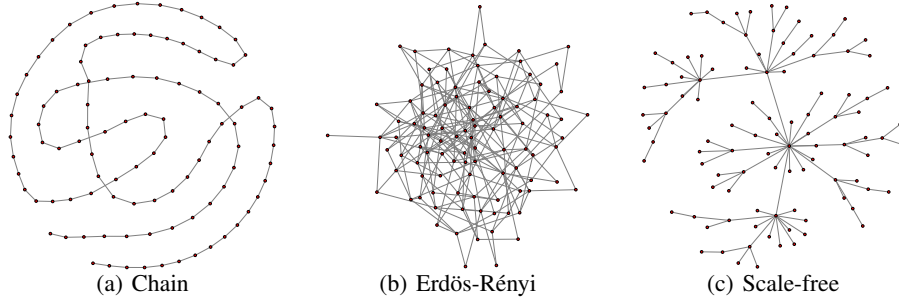

<div align="center">(a) Chain      (b) Erdös-Rényi      (c) Scale-free</div>

Figure 1: Three different graph patterns. To ease the visualization, we choose $d = 100$.

We then generate $n = 200$ independent samples from the t-distribution[1] with 5 degrees of freedom, mean $\mathbf{0}$ and covariance $\mathbf{\Sigma} = \mathbf{\Omega}^{-1}$. For the EPIC estimator, we set $c = 0.5$ in (3.1). For the Catoni's M-estimator, we set $K_{\max} = 10^2$.

To evaluate the performance in parameter estimation, we repeatedly split the data into a training set of $n_1 = 160$ samples and a validation set of $n_2 = 40$ samples for 10 times. We tune $\lambda$ over a refined grid, then the selected optimal regularization parameter is

$$\lambda = \underset{\lambda}{\operatorname{argmin}} \sum_{k=1}^{10} ||\widehat{\mathbf{\Omega}}^{(\lambda,k)}\widehat{\mathbf{\Sigma}}^{(k)} - \mathbf{I}||_{\max},$$

where $\widehat{\mathbf{\Omega}}^{(\lambda,k)}$ denotes the estimated precision matrix using the regularization parameter $\lambda$ and the training set in the $k^{\text{th}}$ split, and $\widehat{\mathbf{\Sigma}}^{(k)}$ denotes the estimated covariance matrix using the validation set in the $k^{\text{th}}$ split. Table 1 summarizes our experimental results averaged over 200 simulations. We see that EPIC outperforms the competing estimators throughout all settings.

To evaluate the performance in model selection, we calculate the ROC curve of each obtained regularization path. Figure 2 summarizes ROC curves of all methods averaged over 200 simulations. We see that EPIC also outperforms the competing estimators throughout all settings.

## 6    Real Data Example

To illustrate the effectiveness of the proposed EPIC method, we adopt the breast cancer data[2], which is analyzed in Hess et al. (2006). The data set contains 133 subjects with 22,283 gene expression levels. Among the 133 subjects, 99 have achieved residual disease (RD) and the remaining 34 have achieved pathological complete response (pCR). Existing results have shown that the pCR subjects have higher chance of cancer-free survival in the long term than the RD subject. Thus we are interested in studying the response states of patients (with RD or pCR) to neoadjuvant (preoperative) chemotherapy.

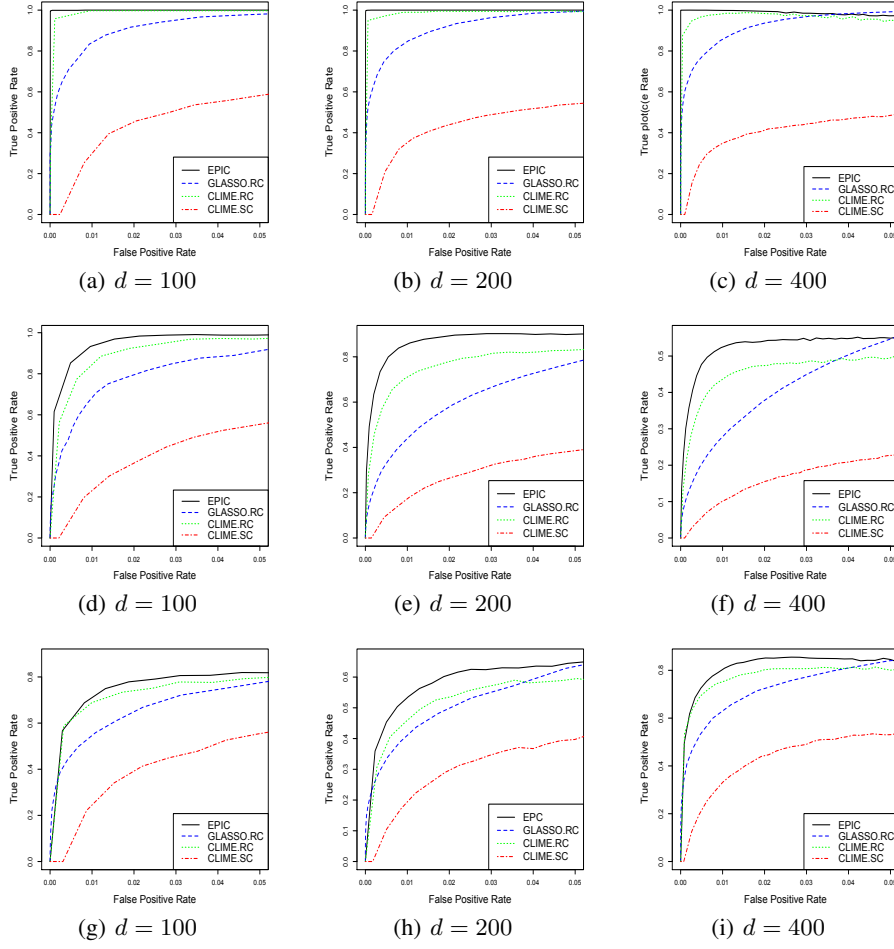

Figure 2: Average ROC curves of different methods on the chain (a-c), Erdös-Rényi (d-e), and scale-free (f-h) models. We can see that EPIC uniformly outperforms the competing estimators throughout all settings.

We randomly divide the data into a training set of 83 RD and 29 pCR subjects, and a testing set of the remaining 16 RD and 5 pCR subjects. Then by conducting a Wilcoxon test between two categories for each gene, we further reduce the dimension by choosing the 113 most signcant genes with the smallest p-values. We assume that the gene expression data in each category is elliptical distributed, and the two categories have the same covariance matrix $\mathbf{\Sigma}$ but different means $\boldsymbol{\mu}^{(k)}$, where $k = 0$ for RD and $k = 1$ for pCR. In Cai et al. (2011), the sample mean is adopted to estimate $\boldsymbol{\mu}^{(k)}$'s, and CLIME.RC is adopted to estimate $\mathbf{\Omega} = \mathbf{\Sigma}^{-1}$. In contrast, we adopt the Catoni's M-estimator to estimate $\boldsymbol{\mu}_k$'s, and EPIC is adopted to estimate $\mathbf{\Omega}$. We classify a sample $\boldsymbol{x}$ to pCR if

$$\left(\boldsymbol{x} - \frac{\widehat{\boldsymbol{\mu}}^{(1)} + \widehat{\boldsymbol{\mu}}^{(0)}}{2}\right)^T \widehat{\mathbf{\Omega}} \left(\widehat{\boldsymbol{\mu}}^{(1)} - \widehat{\boldsymbol{\mu}}^{(0)}\right) \geq 0,$$

and to RD otherwise. We use the testing set to evaluate the performance of CLIME.RC and EPIC. For the tuning parameter selection, we use a 5-fold cross validation on the training data to pick $\lambda$ with the minimum classification error rate.

To evaluate the classification performance, we use the criteria of specificity, sensitivity, and Mathews Correlation Coefficient (MCC). More specifically, let $y_i$'s and $\widehat{y}_i$'s be true labels and predicted labels

Table 1: Quantitive comparison of EPIC, GLASSO.RC, CLIME.RC, and CLIME.SC on the chain, Erdös-Rényi, and scale-free models. We see that EPIC outperforms the competing estimators throughout all settings.

| Spectral Norm: $||\widehat{\boldsymbol{\Omega}} - \boldsymbol{\Omega}||_2$ | | | | | |
|---|---|---|---|---|---|
| Model | d | EPIC | GLASSO.RC | CLIME.RC | CLIME.SC |
| Chain | 100 | 0.8405(0.1247) | 1.1880(0.1003) | 0.9337(0.5389) | 3.2991(0.0512) |
| | 200 | 0.9147(0.1009) | 1.3433(0.0870) | 1.0716(0.4939) | 3.7303(0.4477) |
| | 400 | 1.0058(0.1231) | 1.4842(0.0760) | 1.3567(0.3706) | 3.8462(0.4827) |
| Erdös-Rényi | 100 | 0.9846(0.0970) | 1.6037(0.2289) | 1.6885(0.1704) | 3.7158(0.0663) |
| | 200 | 1.1944(0.0704) | 1.6105(0.0680) | 1.7507(0.0389) | 3.5209(0.0419) |
| | 400 | 1.9010(0.0462) | 2.2613(0.1133) | 2.6884(0.5988) | 4.1342(0.1079) |
| Scale-free | 100 | 0.9779(0.1379) | 1.6619(0.1553) | 2.1327(0.0986) | 3.4548(0.0513) |
| | 200 | 2.9278(0.3367) | 4.0882(0.0962) | 4.5820(0.0604) | 8.8904(0.0575) |
| | 400 | 1.1816(0.1201) | 1.8304(0.0710) | 2.1191(0.0629) | 3.4249(0.0849) |

| Frobenius Norm: $||\widehat{\boldsymbol{\Omega}} - \boldsymbol{\Omega}||_{\mathrm{F}}$ | | | | | |
|---|---|---|---|---|---|
| Model | d | EPIC | GLASSO.RC | CLIME.RC | CLIME.SC |
| Chain | 100 | 3.3108(0.1521) | 4.5664(0.1034) | 3.4406(0.4319) | 16.282(0.1346) |
| | 200 | 5.0309(0.1833) | 7.2154(0.0831) | 5.4776(0.2586) | 23.403(0.2727) |
| | 400 | 7.5134(0.1205) | 11.300(0.1851) | 7.8357(1.2217) | 33.504(0.1341) |
| Erdös-Rényi | 100 | 3.5122(0.0796) | 3.9600(0.1459) | 4.4212(0.1065) | 13.734(0.0629) |
| | 200 | 6.3000(0.0868) | 7.3385(0.0994) | 7.3501(0.1589) | 20.151(0.1899) |
| | 400 | 11.489(0.0858) | 12.594(0.1633) | 13.026(0.4124) | 30.030(0.1289) |
| Scale-free | 100 | 2.6369(0.1125) | 3.1154(0.1001) | 3.1363(0.1014) | 10.717(0.0844) |
| | 200 | 4.1280(0.1389) | 7.7543(0.0934) | 7.8916(0.0556) | 16.370(0.1490) |
| | 400 | 5.3440(0.0511) | 6.3741(0.0723) | 5.7643(0.0625) | 20.687(0.1373) |

of the testing samples, we define

$$\text{Specificity} = \frac{\text{TN}}{\text{TN} + \text{FP}}, \ \text{Sensitivity} = \frac{\text{TP}}{\text{TP} + \text{FN}},$$
$$\text{MCC} = \frac{\text{TPTN} - \text{FPFN}}{\sqrt{(\text{TP} + \text{FP})(\text{TP} + \text{FN})(\text{TN} + \text{FP})(\text{TN} + \text{FN})}},$$

where

$$\text{TP} = \sum_i I(\widehat{y}_i = y_i = 1), \ \text{FP} = \sum_i I(\widehat{y}_i = 0, \ y_i = 1)$$
$$\text{TN} = \sum_i I(\widehat{y}_i = y_i = 0), \ \text{FN} = \sum_i I(\widehat{y}_i = 1, \ y_i = 0).$$

Table 2 summarizes the performance of both methods over 100 replications. We see that EPIC outperforms CLIME.RC on the specificity. The overall classification performance measured by MCC shows that EPIC has a 4% improvement over CLIME.RC.

Table 2: Quantitive comparison of EPIC and CLIME.RC in the breast cancer data analysis.

| Method | Specificity | Sensitivity | MCC |
|---|---|---|---|
| CLIME.RC | 0.7412(0.0131) | 0.7911(0.0251) | 0.4905(0.0288) |
| EPIC | 0.7935(0.0211) | 0.8087(0.0324) | 0.5301(0.0375) |

## Footnotes

[1]The marginal variances of the distribution vary from 0.5 to 2.

[2]Available at http://bioinformatics.mdanderson.org/.

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
