[Reviews · NeurIPS 2013]

Submitted by Assigned_Reviewer_6

A new approach to estimating sparse inverse covariance matrices is presented. The main conceptual idea is to base the estimation on the correlation matrix (instead on the covariance) which makes it possible to use robust rank-based correlation estimators (this is the same idea used in related techniques based on semi-parametric Gaussian copulas). Contrary to graph-lasso type estimators, this version uses a column-wise estimation process and allows for calibration in every such column estimate. The proposed correlation estimator is based on Kendal's tau, and the presented D2P approximation algorithm seems to be scalable and able to exploit structural properties of the problem. Theoretic analysis shows that the estimator achieves the parametric rate of convergence for parameter estimation and model selection.

This is a highly interesting paper that combines sound statistical analysis with convincing experimental evaluation. One question, however, remains: is there any guaranteed that the estimated correlation matrix is positive definite? (I would be surprised....), and if this should not be the case, isn't this a severe technical problem?
Summary: Interesting paper, both on the theoretical and experimental side

Submitted by Assigned_Reviewer_7

In most theoretical works on the estimation of a sparse variance-covariance matrix, it is assumed that the distribution of the observations is Gaussian or sub-Gaussian. Here, the authors assume that the distribution is elliptical. This includes Gaussian distributions, but heavy-tailed distributions as well.

Thanks to the robust estimator proposed by Catoni (Annales de l'IHP, 2012), they propose an estimator and give a rate of convergence. This rate is an improvement on the previous results by Cai et al (JASA, 2011). They describe the algorithm to compute the estimator in detail, and conclude with a nice simulation study.

This is a really strong paper. I would have like some comments on the optimality of the rates obtained in Theorem 4.2.
Summary: This is a really strong paper. I would have like some comments on the optimality of the rates obtained in Theorem 4.2.

Submitted by Assigned_Reviewer_8

Summary of the paper


This paper presents a semi-parametric tuning-free procedure for estimating sparse concentration matrices. This method is applicable to the elliptical distribution family, while most of its competitors only apply to the sub-Gaussian distribution family. The procedure, called ALICE, learns the precision matrix column by column in a similar fashion than the CLIME (Cai et al, 2011), yet with important modifications: a first step is designed to learn the correlation matrix and the associated variances/standard deviations by means of the Kendall's Tau statistic as proposed in Liu et al, 2012. Then, the standard deviations are estimated through a recent proposal of Catoni (2012). In the second step, the inverse correlation is recovered by plugin-in the correlation estimated in the first step in a convex program similar to the CLIME, yet with a modification that allows for some calibration between the columns. This leads to the tuning-free property of the ALICE. Finally, the third step recovers the inverse covariance matrix after a simple rescaling of the inverse correlation matrix. A dual inexact iterative projection algorithm is given to (approximately) solve the convex program in step 2. Theoretical guarantees, in both terms of estimation consistency and selection consistency are provided, equivalent to those proposed for the CLIME estimator. A simulation study on synthetic and breast cancer data illustrates the good performance of the method compared to the state-of-the-art methods.

Comments

The introductory part in this work gives a concise and appropriate bibliography. Motivations for semi-parametric estimators and limitations of the state-of-the art methods are clearly introduced. Contributions of the current proposal (ALICE) are clearly stated.

When the method (part 3) is presented though, some additional justification and explanation could have helped the reading and the general understanding: why choosing Kendall's Tau + Catoni's M estimator? Any other possibilities? It is also not very clear what is due to the authors: using Kendall's Tau for correlation estimation in the elliptical distribution framework is apparently due to a previous work (Liu et al, 2012b): what of Catoni's M estimator? Is this already used in such a context?

The convex program solved looks like a modification of the CLIME criterion. Some more connexion at this stage of the paper (part 3.2) with the CLIME would have been appropriate. As a matter of fact, it would have been easier to judge the originality of the algorithm that follows, whose presentation is somewhat cumbersome. The algorithm gives an approximation of the target estimator, but no quantification are made on this approximation. Providing a global complexity of the procedure would have been a good idea, too.

The same kind of remarks go for the theoretical part: though quite nice, the properties derived by the authors are close to the properties existing for the CLIME; and it is hard to evaluate how straightforward it can be deduced from the work of Cai et al.

Regarding the numerical experiments, the good behavior of ALICE on non strictly Gaussian data is nicely illustrated. Still, I do not understand why the proposal of Liu et al., 2012b, is not included in the analysis. The same for the GLASSO with a prior transformation of the sample covariance matrix, as implemented in the huge package with the non paranormal transformation.

It is a pity that no illustration is made of the tuning-free property of ALICE, since the calibration parameters are tuned with cross-validation in the numerical experiments.
Summary: A paper that provides a serious, comprehensive and apparently new proposal for sparse precision matrices inference in a wider settings than the usual Gaussian world.
The writing is ok, although some efforts could still be made regarding the presentation of the method. I have concerns regarding the numerical comparison and connexion to the existing state-of-the art methods.
Author Feedback

Author rebuttal: Reviewer 6#. Our procedure does not enforce the solution to be positive definite. This does not cause trouble for LDA.

Reviewer 7#. The rates of convergence under the matrix L1 and spectral norms are minimax optimal over the defined model class.

Reviewer 8#. The combination of Cantoni’s M-estimator and Kendall’s tau correlation matrix estimator allows us to get the optimal scaling and rates of convergence within the elliptical family. For experiments, we considered the CLIME.R estimator, which is a combination of Liu et al. (2012b) and the Cantoni’s M-estimator. The ALICE estimator outperforms the CLIME.R estimator. The overall convergence rate of the D2P algorithm O(1/t) has been established in He et al. (2012).